# Current Progress and Outlook of Nano-Based Hydrogel Dressings for Wound Healing

**DOI:** 10.3390/pharmaceutics15010068

**Published:** 2022-12-26

**Authors:** Xiao Zhang, Pengyu Wei, Zhengyang Yang, Yishan Liu, Kairui Yang, Yuhao Cheng, Hongwei Yao, Zhongtao Zhang

**Affiliations:** 1Department of General Surgery, Beijing Friendship Hospital, Capital Medical University, Beijing 100050, China; 2National Clinical Research Center for Digestive Diseases, Beijing 100050, China; 3Jun Skincare Co., Ltd., Jiangsu Life Science & Technology Innovation Park, Nanjing 210093, China; 4School of Chemistry and Chemical Engineering, Nanjing University, Nanjing 210023, China; 5State Key Laboratory of Pharmaceutical Biotechnology, Medical School and School of Life Sciences, Nanjing University, Nanjing 210093, China

**Keywords:** wound healing, wound dressings, hydrogels, nanotechnology, nanomedicine

## Abstract

Wound dressing is an important tool for wound management. Designing wound dressings by combining various novel materials and drugs to optimize the peri-wound environment and promote wound healing is a novel concept. Hydrogels feature good ductility, high water content, and favorable oxygen transport, which makes them become some of the most promising materials for wound dressings. In addition, nanomaterials exhibit superior biodegradability, biocompatibility, and colloidal stability in wound healing and can play a role in promoting healing through their nanoscale properties or as carriers of other drugs. By combining the advantages of both technologies, several outstanding and efficient wound dressings have been developed. In this paper, we classify nano-based hydrogel dressings into four categories: hydrogel dressings loaded with a nanoantibacterial drug; hydrogel dressings loaded with oxygen-delivering nanomedicines; hydrogel dressings loaded with nanonucleic acid drugs; and hydrogel dressings loaded with other nanodelivered drugs. The design ideas, advantages, and challenges of these nano-based hydrogel wound dressings are reviewed and analyzed. Finally, we envisaged possible future directions for wound dressings in the context of relevant scientific and technological advances, which we hope will inform further research in wound management.

## 1. Introduction

The skin is the human body’s largest organ, with an adult area of approximately 1.2~2.0 m^2^ and a thickness of approximately 0.5~4 mm [1,2,3]. As the first barrier of the human body to the outside world, intact skin has the roles of feeling external stimuli, regulating body temperature, and protecting the human body from external injuries; as a result, it is also the organ that is most likely to receive injuries [4,5]. Various external stimuli (puncture, scratch, surgery, burns, etc.,) or intrinsic pathologies (diabetes, chronic arterial and venous insufficiency, etc.,) may be triggering factors for skin wounds [6]. The normal healing process of skin wounds is divided into four stages: hemostasis, inflammation, proliferation, and remodeling (Figure 1) [7]. Acute wounds normally proceed through an orderly and timely reparative process that results in sustained restoration of anatomic and functional integrity. However, chronic wounds fail to achieve anatomical and functional integrity timely and orderly due to vascular insufficiency, infection, and microbial proliferation in the wound bed [1,8], which increases the difficulty of wound management and patient pain.

Wound dressings can cover a wound to provide a temporary barrier against infection and prevent tissue dehydration, facilitating a suitable healing environment for the wound. Traditional dressings, such as gauze, cotton pads, and bandages, are widely used in clinical practice because they are inexpensive and easy to use [9,10]. However, they are often too dry and cannot provide an ideal healing environment for wounds. The search for better wound dressings has continued as the understanding of the wound healing process has increased and biomaterials have evolved.

The ideal wound dressing should have the following characteristics: (1) good tissue compatibility, nontoxicity, and harmlessness; (2) good moisturizing properties; (3) sufficient mechanical strength; and (4) appropriate surface microstructure and biochemical properties to promote cell adhesion, proliferation, and differentiation [9]. In recent years, many kinds of wound dressings have been successfully developed. An increasing number of studies have also focused on loading nanomedicines on hydrogel dressings as a wound healing treatment method by combining the two materials and taking advantage of both to achieve better wound healing promotion. In this paper, we summarize and discuss different types of hydrogel dressings loaded with nanomedicines to provide a reference and direction for the exploration of wound dressings.

## 2. Hydrogels and Nanomaterials for Wound Healing

Among the various materials, hydrogels have been developed for use in a variety of medical fields [11,12]. They can mimic the microstructure of the extracellular matrix (ECM) due to their three-dimensional hydrophilic network and feature good ductility, high water content, and favorable oxygen transport [5,13,14]. Hydrogels are mostly fabricated by physical or chemical cross-linking of various hydrophilic polymers. Depending on the source of materials, hydrogels can be divided into natural polymer-based hydrogels, synthetic polymer-based hydrogels, biomimetic polymer-based hydrogels, and hybrid hydrogels [15,16]. The property of hydrogels is also affected by different cross-linking strategies. For instance, chemically cross-linked hydrogels generally feature better stability, while physically cross-linked hydrogels tend to be safer when applied in vivo [17,18]. To expand the potential applications of hydrogels in wound healing, researchers have also developed multifunctional hydrogels to improve their performance [19].

Hydrogels play an important role in the field of wound treatment. First, hydrogels keep the wound environment moist and reduce the pain of patients owing to their high water absorption and swelling properties. Second, hydrogel dressings are of good elasticity and toughness so that they can form a perfect barrier around the wound during the healing process. Third, they can be used as carriers of various drugs and bioactive substances to accelerate wound healing [19]. In addition, hydrogel dressings can be designed based on specific needs, which opens up countless possibilities for their application [5,15,20]. For instance, although polymeric materials are ineffective at preventing mechanical damage, self-healing hydrogels can intrinsically automatically heal damages and restore themselves to normality in time, which will be helpful to improve their performances in different application fields [21]. Some researchers also introduced the fabrication of self-adapting hydrogels, which can automatically change shape without external stimuli. This fascinating feature enables self-adapting hydrogels as an excellent drug carrier for in vivo wound treatment [22]. For the past few years, smart hydrogels could attract attention as a promising material in varieties of fields. They are termed “smart” because of their response to specific physical and chemical environmental stimuli [19]. In addition, hydrogels can be designed as cell-delivery systems based on specific tissue properties [23].

Nanomaterials are materials that have structural components smaller than 100 nm in at least one dimension [24,25]. Their special properties, such as nanoscale size and high surface area to volume ratio, have led to their rapid development and proven potential in recent years in medical fields, including drug and gene delivery and biosensor applications [26,27]. Compared with other materials, nanomaterials exhibit superior biodegradability, biocompatibility, and colloidal stability in wound healing and can play a role in promoting healing through their nanoscale properties or as carriers of other drugs [1,28,29], making them a good material choice in the field of wound healing. Nanomaterials used in wound dressings comprise designed metal-based nanoparticles and biomaterials that offer an unmatched approach to accelerate wound repair and the tissue-remodeling process [30]. Silver nanoparticles(AgNPs), one of the most extensively studied metallic nanoparticles, are becoming the potential candidate of choice for wound repair due to their unique anti-inflammatory properties and antibacterial activity [31]. In addition to metal nanoparticles, other nanomaterials, including porous silicon nanoparticles (pSi NPs) [32], chitosan nanoparticles [33], and other nanocarriers, also have great promise for clinical use.

With the ongoing exploration and development of wound dressings, hydrogels are becoming good carriers for different nanomaterials to accelerate the wound healing process. Many kinds of hydrogel dressings loaded with nanomedicines have been developed (Table 1). In general, nanomedicines loaded by hydrogels can be divided into nucleic acid nanomedicines, oxygen-delivery nanomedicines, antibacterial nanomedicines, and other nanodelivered drugs which play different roles in the wound healing process.

## 3. Hydrogel Dressing Loaded with Nanomedicines

### 3.1. Hydrogel Dressing Loaded with Nanoantibacterial Drug

The skin is a barrier against pathogenic microorganisms, and when the skin is damaged, external microorganisms can colonize and grow within the wound, causing a prolonged healing time. Bacteria and fungi can even invade the body and cause serious infections in cases where the body’s immunity is low. Prevention of microbial infections is one of the basic requirements of the wound healing process. Many antimicrobial dressings have been used to inhibit microbial infections in wounds [34]. However, traditional wound dressings have disadvantages such as easy adhesion to the wound, poor barrier effect and poor hemostasis [50]. In recent years, laboratories have devoted themselves to developing new antimicrobial materials, including metal or metal oxide nanoparticles, metal-organic networks, nanoenzymes, cationic polymers, dendritic polymeric peptides, carbon-based nanostructures, nanocellulose-based materials, and supramolecular complexes [51], which exhibit better antimicrobial effects.

Silver nanoparticles (AgNPs) are widely used as an effective antimicrobial agent in wound dressings. The antimicrobial mechanism of AgNPs can be explained in various ways. Most researchers believe that AgNPs exert their antimicrobial effects by releasing Ag^+^ [52,53]. Since Ag^+^ is positively charged and has a small diameter, it has a large specific surface area and can adsorb on the surface of negatively charged bacteria and penetrate their cell walls, leading to bacterial rupture, causing bacterial death, and preventing their reproduction. Some scholars believe that AgNPs can induce the expression of free radicals and thus oxidize the outer membrane of the bacterium, which in turn causes bacterial lysis and death [54]. It has also been suggested that AgNPs may achieve antibacterial effects by affecting bacterial signaling pathways [55]. In conclusion, as an inorganic antibacterial agent, AgNPs have the advantages of broad antibacterial enhancement, high antibacterial efficiency, and long action time. Xiao et al. (2021) synthesized a conductive polymer-based hydrogel system (CPH) using polyvinyl alcohol (PVA) and gelatin as the main matrix materials [56]. They then loaded Ag NPs on CPH by soaking and fabricated a medical gel applicator (Ag NPs/CPH) for severe wound infections (Figure 2).

Chitosan (CS) is a natural polysaccharide formed by the acetylation of chitin and consists of d-glucosamine and n-acetylglucosamine [57,58]. CS exerts its antibacterial activity through the following mechanisms: chitosan with high molecular weight could form a dense layer on the cell surface, prevent nutrient uptake, and interfere with the bacterial metabolism by electrostatic accumulation; small molecular weight chitosan can penetrate membranes and block the transcription of RNA by embedding in the deoxyribonucleic acid chain; In addition, chitosan can chelate metal ions and essential nutrients that are important for the growth of the microbial cell [59,60,61,62,63]. In addition to its good antibacterial properties, CS has excellent water absorption ability, biocompatibility, and degradability. In addition, as a good natural hemostatic material, CS can mediate red blood cell aggregation and repair of damaged tissues [64,65]. Owing to these many advantages, CS has developed into a commonly used trauma dressing. However, the antimicrobial ability of CS is not yet sufficient to prevent microbial infections in wounds [58]; therefore, CS is often used as a support material for loading antimicrobial drugs in the preparation of wound dressings.

Zhou et al. (2021) developed a chitosan composite sponge dressing loaded with iturin-AgNPs [34]. Iturin is a cyclic structure consisting of seven amino acids and a 13–19C β-hydroxy fatty acid. Iturin-AgNP complexes have good antibacterial effects against a variety of bacteria and fungi and do not cause drug resistance while reducing the Ag content, which has potential application in wound healing. The addition of iturin-AgNPs significantly improves the antimicrobial activity of CS dressings and has great potential for wound care applications. Zhang et al. (2021) designed a bimolecular layer hydrogel wound dressing [66]. The upper layer was chitosan nanoparticles loaded with Ag@MOF (metal-organic frameworks), and the lower layer was PACS (PVA polyvinyl alcohol/SA sodium alginate/CS chitosan) hydrogel prepared by a freeze-thaw process. The upper layer (Ag@MOF/CSNPs) has good antibacterial activity and can inhibit microbial invasion, while the lower layer (PACS) has a uniform pore distribution, good water absorption ability, swelling capacity, oxygen permeability, and biocompatibility and it is used to promote epithelial tissue growth. The results of in vivo experiments showed that this bilayer dressing can promote re-epithelialization and reduce the inflammatory response, making it an ideal dressing for accelerating wound healing.

Reactive oxygen species (ROS) are broad-spectrum bactericides that can kill bacterial propagules, bacterial spores, viruses, and fungi and have good killing effects on protozoa and their oocysts, as well as destroying bacterial toxins and hepatitis B surface antigens. Wound dressings based on reactive oxygen species’ antimicrobial activity have also been widely reported. Liu et al. (2018) produced a catechol-modified chitosan membrane that catalyzed the transfer of electrons from the physiological reducing agent ascorbic acid to O_2_ for sustained ROS production and provided ascorbic acid-dependent antimicrobial activity [67]. In vitro antimicrobial experiments showed that catechol-chitosan membranes inhibited bacterial growth and alleviated incisional infections in the reduced state, and this material is expected to provide a new solution for wound management. Xie et al. (2022) reported an alloy nanostructure, metal-phenolic nanoplatform (Ag@Cu-MPN_NC_), in which the Cu structural domain of the nanostructure can cause an increase in ROS levels, while the Ag structural domain can simultaneously disrupt bacterial cell membranes, allowing ROS to effectively penetrate the bacterial cytoplasm and oxidize intracellular proteins, further enhancing its bactericidal effect [51]. In an in vivo model of infected rats, Ag@Cu-MPN_NC_ effectively killed bacteria, promoted hematopoietic reconstruction, and accelerated wound healing without adverse effects. The coating is highly compatible with the current widely used wound dressing matrices and has good application prospects.

Adjusting the wound pH from alkaline to acidic is a simple and effective way to reduce microbial colonization and infection in wounds. In addition, an acidic environment can inhibit protein hydrolase activity, increases cellular oxygenation, and promotes fibroblast growth and neovascularization, all of which can contribute to wound healing. Piva et al. (2018) proposed an agarose membrane containing the nanobacterial substance Cs_2.5_H_0.5_PW_12_O_40_ as an efficient proton delivery agent that reduces the surface pH of nanocomposites to the range of 7.0 > pH ≥ 3.0 [35]. The nanocomposite membranes containing 20 wt % Cs_2.5_H_0.5_PW_12_O_40_ NPs had the highest antimicrobial activity at a pH of 3.0 on the surface. Its broad antimicrobial effect has been demonstrated in *Escherichia coli*, *Staphylococcus aureus*, *Candida albicans*, and *Aspergillus fumigatus*. Li et al. (2021) grafted Fe-MIL-88NH2 nanozyme to glycidyl methacrylate functionalized dialdehyde chitosan via a Schiff base reaction, and acryloyl Pluronic 127 (PF127-DA) was used as a cross-linking agent to fabricate nanozyme composite cryogels (CSG-MX) as a wound dressing [36]. The material also has a local pH-regulating effect, and with its high hydrophilicity, it can achieve rapid fluid absorption and bactericidal effects, providing a practical strategy for anti-infection in wound healing.

Overcoming multidrug-resistant (MDR) infections is a challenge and an urgent need for wound healing. Traditional antimicrobial biomaterials, including inorganic nanomaterials (silver, zinc, copper) and organic molecules (quaternary ammonium salts, alkylated polyethyleneimine), can be used to treat MDR infections [68]; however, the cytocompatibility and hemocompatibility of these antimicrobial biomaterials are poor. Xi et al. (2018) developed an antimicrobial composite peptide-based nanofibre matrix as a multifunctional platform to inhibit MDR and promote wound healing [69]. The composite nanofibre consists of poly(citrate)-ε-polylysine (PCE) and polycaprolactone (PCL) and has a tensile elastic modulus similar to that of human skin tissue as well as excellent hydrophilic properties. Bacterial cell membranes are easily disrupted due to the neutralization of PCE (positive charge) and bacterial cell membranes (negative charge). The PCL-30% PCE nanofibre matrix has efficient antibacterial activity against *E. coli*, *Pseudomonas aeruginosa*, *Staphylococcus aureus*, and methicillin-resistant *Staphylococcus aureus* (MRSA) while maintaining good cytocompatibility and hemocompatibility. It effectively prevents MDR bacterial-derived wound infections. As a multifunctional dressing, the PCL-30% PCE hybrid nanofibre matrix has great potential in promoting chronic wound healing and skin tissue regeneration by stimulating the formation of epidermal, dermal, and follicular tissues. Liu et al. (2020) used amphiphilic, oxadiazole group-modified quaternary ammonium salt (QAS)-conjugated poly(ε-caprolactone)-poly(ethylene glycol)-poly(ε-capro-lactone) micellar nanoantimicrobial agent (pcec-QAS) [37]. An antimicrobial bioresorbable hydrogel was developed for skin wound healing of MRSA infections. The hydrogel showed broad antibacterial activity against MRSA, *Escherichia coli* and vancomycin-resistant *Staphylococcus* and promoted cell spreading, proliferation, and migration without cytotoxicity, with promising applications.

Unlike traditional chemical sterilization strategies, photothermal therapy (PTT) converts light energy into local physical heat, which has the advantages of broad-spectrum antimicrobial activity, noninvasiveness, and deep tissue penetration. Various types of light-absorbing materials have been explored for in vivo biomedical applications, including gold-based nanomaterials, carbon-based nanomaterials, CuS [70], and other inorganic photothermal agents and organic photothermal agents such as indocyanine green (ICG), porphyrins [71,72], and dopamine (PDA) [38,73]. Ding et al. (2021) fabricated a dressing for chronic wounds called Au-EGCG@H, which fuses Au-EGCG into the hydrogel [39]. Au-EGCG, the novel gold cage (AuNCs) modified with epigallocatechin gallate (EGCG), has a high and stable photothermal conversion efficiency under near-infrared irradiation(NIR), and it can produce plenty of reactive oxygen species (ROS), inducing bacterial lysis and apoptosis(Figure 3). Their further experiments verified the effectiveness and biocompatibility of this dressing. Dopamine nanoparticles (PDA NSs) are promising materials with high biocompatibility, mussel-inspired adhesive characteristics, and excellent photothermal conversion efficiency. To date, there have been few reports of antimicrobial hydrogel dressings loaded in PDA NSs. Zeng et al. (2021) introduced PDA NPs into XK (consisting of xanthan gum and konjac dextran), a food gum matrix for skin wound healing, and then developed a nanocomposite hydrogel, XKP [73]. This material has broad-spectrum bactericidal activity and does not cause bacterial resistance. In addition, the XKP hydrogel has good elasticity and adjustable water absorption ability, allowing it to adapt to the shape of the wound and provide a suitable moist environment. This strategy provides further options for the clinical selection of suitable wound healing materials. Liu et al. (2021) concluded that the application of the PTT sterilization strategy alone is not sufficient to achieve the desired therapeutic effect, and the high temperature may cause damage to the surrounding tissues [38]. Considering the antibacterial effect of NO on a variety of bacteria, including Gram-positive and Gram-negative bacteria, as well as the possibility of promoting wound healing through various mechanisms, such as increasing myofibroblasts, promoting wound contraction, and collagen deposition. Liu et al. used a strategy of PTT combined with gas therapy to combine NO donors (N,N’-di-sec-butyl-N,N’-dinitroso-1,4-phenylenediamine, BNN6) onto the surface of 2D PDA to obtain a PDA-BNN6 nanocomposite with good photothermal effect and NO release function. The PDA-BNN6 NS nanocomposites were further physically mixed with an injectable hydrogel composed of adipic acid dihydrazide-modified γ-polyglutamic acid (γ-PGA-ADH) and aldehyde-(F127-CHO), which acted as an antibacterial wound dressing for full-thickness skin wound healing. This hydrogel system has significant potential for clinical applications in wound anti-infection and wound healing.

### 3.2. Hydrogel Dressing Loaded with Oxygen-Delivering Nanomedicines

The wound healing process is inseparable from bioenergy consumption (e.g., adenosine triphosphate, AT), and oxygen is necessary for bioenergy production by participating in the tricarboxylic acid cycle, fatty acid oxidation, etc. Adequate oxygen ensures normal cellular function, thus promoting wound contraction, avoiding inflammation, increasing differentiation of keratinocytes, and migration during wound healing [74,75]. The normal wound healing process will be affected if a wound is hypoxic for various reasons (e.g., old age, diabetes, etc.,) [76]. Hyperbaric oxygen therapy is an adjunctive therapy that promotes wound healing by increasing the oxygen supply to the peri-wound tissues. However, the process of hyperbaric oxygen therapy requires the patient to be placed in a specific environment with 100% oxygen inhalation, which may not only increase the cost of treatment but also produce short-term myopia worsening, claustrophobia, oxygen toxicity, and other adverse effects [75]. Some current studies have used hydrogel-loaded nanomaterials for local oxygen supply around wounds, achieving better results in promoting wound healing and offering new possibilities for wound care.

During the exploration of local oxygen therapy for wounds, peroxide is a relatively common raw material for oxygen supply. Shiekh et al. first incorporated calcium peroxide into the elastomeric antioxidant polyurethane (PUAO) to fabricate polyurethane-based oxygen-releasing antioxidant scaffolds (PUAO-CPO), which can release oxygen continuously for more than 10 days. In addition, adipose-derived stem cell (ADSC) exosomes are cell-derived nanovesicles that carry growth factors and microRNAs. They can modulate wound healing and angiogenesis process by stimulating cell migration and proliferation [77]. Therefore, they created exosome-laden oxygen-releasing antioxidant wound dressing OxOB by appending adipose-derived stem cell (ADSC) exosomes to PUAO-CPOs. This dressing combines the benefits of multiple materials, including providing a matrix for cell migration, attenuating oxidative stress, and providing sustained oxygen to the wound [78].

The hyperbaric oxygen-generating (HOG) hydrogels (HOG-gels) reported by Park et al. can maintain high oxygen levels in vitro for up to 12 days. In their design process, they mediated the oxidative cascade reaction of GtnSH through calcium peroxide to generate oxygen and form a hydrogel network in situ. They also found that the oxygen release behavior can be effectively controlled by varying the amount of calcium peroxide, which provides a broader prospect for the application of HOG gel [79]. In addition, Zehra et al. (2020) fabricated PCL (polycaprolactone)-based oxygen-releasing wound dressings by using peroxide-based oxygen-producing materials [40]. The oxygen-generating wound dressing designed by Fatemeh et al. (2021) was composed of H_2_O_2_-loaded polylactic acid (PLA) microparticles embedded within a chitosan/β-glycerophosphate (β-GP) thermosensitive hydrogel covered with a layer of decellularized human-amniotic membrane(AM) [80].

In addition to the use of peroxides for oxygen production through different techniques, scientists are constantly searching for other oxygenating materials that can be used in wound dressings. Nanooxygenated (NOX) powder, which is perfluorodecalin-encapsulated albumin nanoparticles, is a safe lyophilized nano additive for dissolving and delivering oxygen [81,82]. Yang et al. mixed NOX powder into hyaluronate gel to form a NOX gel and made a wound dressing. They verified the superior oxygenation and wound healing effects of this oxygenated wound dressing in a murine acute wound model and a diabetic chronic wound model. Its better preservation and transportation advantages may mean better clinical application prospects [41]. Considering that photosynthesis is a common mode of oxygen production, Chen et al. (2020) found that wound dressings made from 1-mm-diameter hydrogel beads containing active *Synechococcus elongatus* (*S. elongatus*) PCC7942 and carbonates (CO_3_^2−^ and HCO_3_^−^) can effectively provide topical dissolved oxygen (TDO), which is more efficient than topical gaseous oxygen (TGO) penetrating the skin [83] and better promotes wound healing [42].

### 3.3. Hydrogel Dressing Loaded with A Nanonucleic Acid Drug

In recent years, with the increasing understanding of the wound healing process, nucleic acid therapies that enhance or inhibit different signaling pathways at the genetic level have become a new idea in wound therapy with the advantages of “longer duration of action, higher specificity of action, higher target selectivity, and software-designable sequences” [84,85]. Gene therapy for wounds mainly refers to therapeutic nucleic acids (including small interfering RNA (siRNA), oligonucleotides, plasmid DNA, antisense oligonucleotides (ASO), microRNA (miRNA) mimics, anti-miRNA oligonucleotide (AMO), aptamers, and messenger RNA (mRNA)) that regulate cell motility, angiogenesis, epithelialization, and oxidative stress through different signaling pathways in various steps of wound healing to achieve a more efficient promotion of wound healing [86,87,88].

However, the development of nucleic acid drugs in the field of wound healing treatment has not been smooth. There are still many challenges in the research process of these drugs: (1) Their instability makes them vulnerable to degradation by nucleases, with a subsequent loss of function; (2) it is difficult for nucleic acid molecules to enter the cytoplasm or nucleus to play therapeutic roles because of their size and other specific characteristics; (3) when applied by injection, nucleic acids may also induce immune reactions as exogenous substances, which will affect the safety of the drug; (4) they will also increase patient pain because of their need to be injected repeatedly to maintain the effect; and (5) repeated injections may cause drug accumulation and increase the burden on the liver [85,89,90]. To solve these problems, researchers have found that some nanocarriers can improve the delivery efficiency and cellular uptake rate of nucleic acids. Through the delivery of hydrogel dressings, these nucleic acid drugs can be released locally and continue to exert lasting effects locally in wounds [43,44,45,46,91,92,93]. Several will be described in the following section.

Matrix metalloproteinase-9 (MMP-9) is a member of the zinc-dependent endopeptidase family and is involved in tissue remodeling. Its levels are elevated in many diseases, including myocardial infarction, stroke, and cancer. In recent years, several research teams have found that MMP-9 plays a role in the degradation of ECM and tissue reconstruction during wound healing [94,95,96], but when overexpressed, it leads to the inactivation of important growth factors and affects granulation tissue and early connective tissue formation [97,98,99]. Therefore, local downregulation of MMP-9 expression in wounds is one of the available targets for nucleic acid therapy. Among the many therapeutic modalities that target MMP9 expression, MMP-9-specific small interfering RNA (siRNA) (siMMP9) can effectively silence MMP9 gene expression due to its precise mode of action [100,101]. To solve the problem of the inaccessibility of siMMP9 to cells due to charge–charge repulsion [102,103], Li et al. (2020) chose to complex siMMP9 with hyperbranched cationic polysaccharide derivatives (HCP) to facilitate its delivery. They encapsulated HCP/siMMP9 in bacterial cellulose (BC) to form a wound dressing, which combined the advantages of BC to make their hydrogel dressing effective in providing an anti-infection barrier, maintaining a moist wound environment with good breathability while slowly releasing siMMP9 in the wound environment for localized and specific inhibition of MMP9 expression [43]. Their team also used glycogen triethylenetetramine (Gly-TETA, GT) as a carrier for siMMP9 following the same principle and encapsulated it in PM hydrogel (the thermosensitive hydrogel made of Pluronic F-127 (PF-127) and methylcellulose (MC)) to achieve similar therapeutic effects (Figure 4) [91].

During the proliferative phase of the four stages of wound healing, a key process is the generation of new blood vessels. In intact tissues, the microvascular system is suspected to deliver sufficient nutrients and oxygen to the tissue and to remove oxygen and carbon dioxide. In the wound setting of patients with diabetes mellitus, peripheral vascular disease, etc., insufficient angiogenesis significantly affects the transport of nutrients needed for the wound healing process and thus delays wound healing [45,104,105]. It has been found that downregulating the expression of the microRNA miR-29a in peri-wound cells can promote angiogenesis and type I collagen synthesis, providing an idea for wound healing treatment [106]. Based on this, Yang et al. (2021) mixed adipic dihydrazide-modified hyaluronic acid (HA-ADH), oxidized hydroxymethyl propyl cellulose (OHMPC), oridonin (ori)-loaded alginate microspheres (Alg@ori), and siRNA-29a gene-loaded hyaluronic acid-polyethyleneimine complex HA-PEI@siRNA-29a (HA-PEI@siRNA-29a), resulting in a novel hydrogel named Gel/Alg@ori/HA-PEI@siRNA-29a to achieve downregulation of miR-29A by slow release of siRNA-29a [44].

Vascular endothelial growth factor (VEGF), best known as an important regulator of angiogenesis during wound healing, promotes wound healing by allowing inflammatory cells to enter the site of injury and stimulating endothelial cell proliferation, migration, and other mechanisms. However, when VEGF is applied directly to a wound, the protease-rich wound environment significantly affects its stability and biological activity. Repeated and frequent injections are required to achieve the desired effect, significantly increasing the complexity of the operation and the cost of treatment [46,92]. To better exploit the effects of VEGF, Wang et al. synthesized complex hydrogels with chemically modified hyaluronic acid (HA), dextrose (Dex), and β-cyclodextrin (β-CD) and then used them to promote burn wound healing by binding resveratrol (Res) and vascular endothelial growth factor (VEGF) plasmids [45]. This dressing integrates VEGF plasmid therapy along with Res, which has been shown to significantly upregulate the conditional expression of VEGF in human skin cells [107,108] and can act synergistically with VEGF plasmids.

Nucleic acids (NAs) have wide prospects for exploration and application in the field of wound healing because of their gene-level regulation, which allows their application in all steps of wound healing to penetrate. Since the decreased expression of miRNA146a was found to correlate with increased expression of proinflammatory factors affecting wound healing [109], Sener’s team cross-linked miRNA146a onto cerium oxide nanoparticles (CNPs) and developed chemical-free amphoteric hydrogels to make them flexible, self-healing, and injectable and to ensure sustained release [46]. Polydeoxyribonucleotides (PDRNs), DNA fragments extracted from the sperm cells of Oncorhynchus mykiss (Salmon trout) or Oncorhynchus keta (Chum Salmon), are reported to have positive therapeutic effects, including increasing collagen synthesis, improving angiogenesis, and promoting cell activity [110,111,112]. Therefore, Jing’s team fabricated a PDRN-loaded CaCO_3_ nanoparticle (PCNP) to improve the delivery efficiency of PDRN and encapsulated it in an alginate/chitosan-based hydrogel to make Gel@PCNPs, which can significantly accelerate wound healing and is a promising wound treatment method [92].

### 3.4. Hydrogel Dressings Loaded with Other Nanodelivered Drugs

As the wound healing process gradually becomes more understood, there are an increasing number of targets for wound healing treatment from which to choose. In addition to the abovementioned nanodelivery of antimicrobial drugs, delivery of oxygen, and delivery of nucleic acid drugs, researchers are also exploring other possible applications of hydrogel dressings.

ROS plays a pivotal role in the normal wound healing procedure. It can regulate the angiogenesis of wound areas and work in defense against infection [113]. However, it is also important to note the detrimental effects of excessive ROS. During wound healing, the prolongation of the early inflammatory time leads to a significant increase in ROS levels. Excess ROS have been shown to promote proinflammatory cytokine expression, oxidative damage, and extracellular matrix (ECM) destruction related to prolonging the wound healing process [114,115,116]. Therefore, the timely removal of reactive oxygen species in the early stages of inflammation is also one of the important methods of wound healing treatment. It has been demonstrated that cerium oxide nanoparticles (CeONs) have great potential to scavenge ROS and play a positive protective role in various ROS overload diseases, such as hepatitis and acute kidney injury [117,118]. Cheng’s team coloaded CeONs with antimicrobial peptides (AMPs) in a sprayable hydrogel wound dressing with both reactive oxygen species scavenging and antimicrobial properties to promote wound healing while effectively and conveniently reducing scar formation [119]. Additionally, Andrabia’s team utilized curcumin, which has anti-inflammatory properties, and coloaded curcumin with CeONs in hydrogels, which achieved considerable antioxidant and anti-inflammatory abilities [47]. Hydrogen molecules are also widely used as novel antioxidants because of their ability to selectively reduce hydroxyl radicals. To prolong the effective reaction time of hydrogen and improve its ability to enter tissues, Chen et al. (2022) reported a hydrogen-producing hydrogel composed of Chlorella and bacteria with sustained hydrogen production up to 60 h [120].

Peptide-based materials are important biomaterials with a variety of structures and functions. In the last decades, self-assembly strategies have been introduced to build peptide-based nanomaterials, which can form well-controlled superstructures with high stability and multivalent effects [121]. Peptide-based hydrogels are biocompatible, and biodegradable and can mimic the extracellular matrix and provide a proper moist environment, which is important for wound healing [122]. Recombinant human PDGF-BB (rhPDGF-BB/becaplermin) has been approved by the FDA for the treatment of diabetic foot ulcers. Santhini et al. (2022) selected the self-assembled peptide RADA 16-I to form a stable nano hydrogel and used it to encapsulate PDGF-BB and demonstrated its angiogenic and wound healing abilities [48]. Mesenchymal stem cells (MSCs) are multipotent adult stem cells that have the potential to differentiate in multiple directions into mesenchymal cell lineages, including adipocytes, osteoblast, chondrocytes, myoblasts, and endothelial cells. Numerous studies have shown that MSCs promote angiogenesis and epithelial cell regeneration, improve granulation, and accelerate wound closure [123]. Xue et al. (2022) applied self-assembled peptide hydrogels and made wound dressings when loading human umbilical cord mesenchymal stem cells (hUC-MSCs), which accelerated skin wound healing by inhibiting inflammation and promoting angiogenesis [49]. In recent years, exosomes (EXO) are considered to be a major contributor to stem cell efficacy [124], which may be attributed to the transfer of cell membrane and cytosolic proteins, lipids, and RNA between cells [125]. The use of EXO-loaded hydrogels as dressings for the treatment of chronic wounds is emerging as a viable option. In a study by Yang et al. (2020), Pluronic 127 (PF-127)-based hydrogel was used with human umbilical cord (hUC)-MSC-EXO for the treatment of diabetic wounds, and the results showed that hUC-MSC-EXO/PF-127 healed more rapidly than the other treatments on days 7, 10, and 14 [126].

## 4. Summary and Prospects

The wound-healing process is complex. Poorly healed wounds, especially chronic wounds, are more difficult to manage and also increase patient expense and pain. Drug delivery through wound dressings allows the dressing to protect the wound and maintain wound wetness while also improving the peri-wound environment and promoting wound healing through the local release of specific drugs or other molecules. In this review, we focus on hydrogel dressings for transporting nanodelivered drugs. Hydrogels with appropriate characteristics (e.g., sprayable, injectable, self-healing, and slow drug release) were selected and designed as dressing materials according to different needs. Researchers have modified or designed drugs through nanotechnology to improve the local availability of drugs, such as increasing drug activity, increasing cell penetration, and improving drug stability. They achieved a variety of local therapeutic effects (such as local antibacterial, oxygen delivery, nucleic acid therapy, and scavenging of reactive oxygen species) and ensured the continuous release of drugs in the wound environment. Researchers have designed an increasing number of nanomedicine-loaded hydrogel dressings based on different principles, providing new options for wound care. ROS plays an essential role in regulating various physiological functions of living organisms. ROS-based nanomedicine is applied to the treatment of various pathological dysfunctions such as bacterial infection, neurodegenerative diseases, cancer, etc., [127]. Though the accurate role of ROS in wound healing is not understood, the control of ROS level is important in this process [113,128]. On the one hand, ROS can attack invading pathogens directly and finally kill them to aid phagocytosis while excessive ROS will damage the surrounding tissue of the wound. On the other hand, a moderate level of ROS can upregulate the production of the vascular endothelial growth factor, which is helpful to accelerate the angiogenesis of the wound. However, excessive ROS will have decelerating effects [129,130,131]. It is clear that the precise balance between low versus the high level of ROS is important to the functional outcome. In this review, we introduced several dressings [47,51,119] that can adjust the level of ROS in the peri-wound environment, but the precise control of ROS level is still a further topic to be explored.

At present, hydrogel dressings still have some drawbacks, such as weak mechanical properties and rapid degradation [132]. When subjected to external forces, hydrogel dressings can crack and lead to bacterial invasion, which can affect the proper functioning of the hydrogel dressing. Therefore, many researchers have designed nanomaterial-based hydrogel dressings with physical or chemical methods to improve the properties of the hydrogel and make it more suitable for wound dressing [45,46,78]. In addition, rats and mice are generally selected as model. However, the wound healing process in rats is different from that in human skin, which may lead to deviations in experimental results. Therefore, the exploration of more suitable animal models is also an important issue for the development of wound dressings [4].

Tissue engineering is currently undergoing rapid development. 3D bioprinting is a novel additive manufacturing technology in this context that enables rapid and precise spatial patterning of cells, growth factors, and biomaterials to create complex three-dimensional tissue structures [133]. In the field of wound dressings, because of the higher accuracy and flexibility of 3D printing technology compared to other production techniques, many researchers have also tried to use this technology to develop various types of wound dressings, such as hydrogels, with some achievements [134,135,136]. This could potentially be a new avenue for wound dressing production in the future.

With the development of the concept of precision medicine and biomaterials science, the requirements for the design of wound dressings are gradually increasing. In addition to the PTTs described in the article, PDTs (photodynamic therapies) have also been discovered, which rely on the interaction of external energy with nanomaterials to generate ROS and thus achieve a broad-spectrum bactericidal effect [137]. Multifunctional photoresponsive hydrogels (MPRHs), which combine the advantages of light and hydrogels, are also increasingly used in wound repair [13]. PTT, PDT, and MPRH therapies are applied to promote wound healing by interacting with external energy to produce antimicrobial effects or promote tissue regeneration. Although there are still many challenges to be overcome before their clinical application, they provide a broader idea for the treatment of wound healing. One of the more interesting topics is how to determine the true state of the wound underdressing coverage and to modify the treatment plan for wound healing promptly. Most hydrogel dressings facilitate the observation of wound status due to their transparent appearance compared to traditional dressings such as gauze and bandages. However, the development of “smart dressings” also offers the possibility to detect and diagnose the wound condition in real-time with greater accuracy by integrating different types of sensors into the dressing and connecting to smart devices to monitor the temperature, pH, ROS levels, etc., of the wound environment in real-time and to regulate the release of relevant drugs within the dressing to facilitate the wound healing process [138,139,140]. Combining the advantages of smart dressings and nanocarrier hydrogel dressings is expected to provide personalized dressing design according to the characteristics of the wound to facilitate wound healing. With the development of smart wearable devices and big data, such wound dressings with both monitoring and therapeutic functions may also provide the possibility of achieving remote wound care, which will facilitate more convenient and accurate wound care for patients with mobility impairments [9].

In conclusion, nano-based hydrogel dressings combine the advantages of nanotechnology and hydrogel dressings to provide a better solution for promoting wound healing. An increasing number of new technologies are also gradually being integrated with the development of wound dressings, which requires more exploration to advance the development of wound care. However, how to safely and effectively translate these new technologies from laboratory development to clinical application remains the key to our further explorations.

## Figures and Tables

**Figure 1 pharmaceutics-15-00068-f001:**
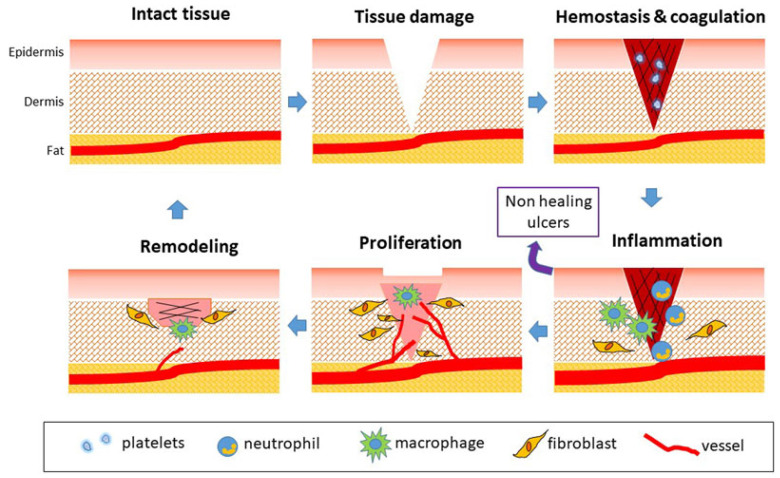
Phases of physiological wound healing. Reproduced with permission [7].

**Figure 2 pharmaceutics-15-00068-f002:**
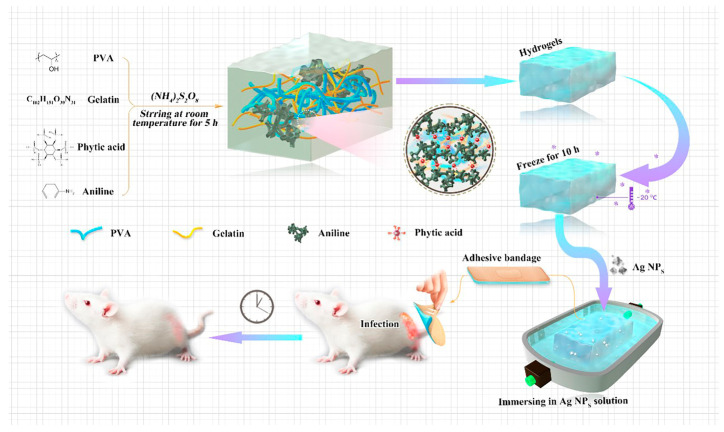
The schematic diagram of the synthesis process of the Ag NPs/CPH, and its applications the animal model. Reproduced with permission [56].

**Figure 3 pharmaceutics-15-00068-f003:**
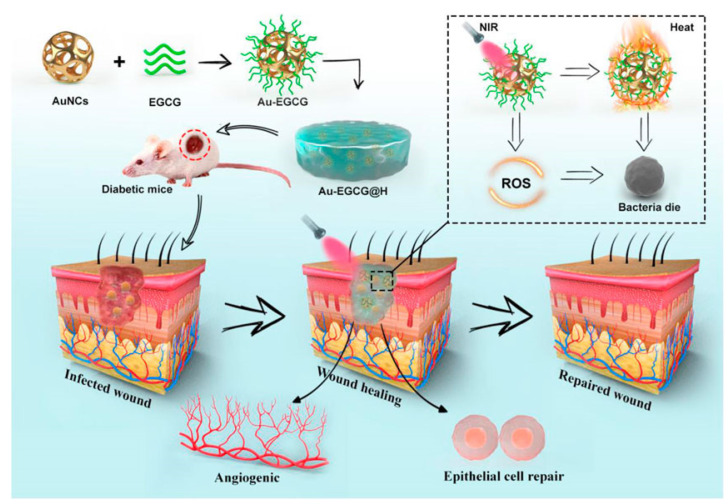
The schematic diagram of the synthesis of Au-EGCG@H nanocomposites and the process of sterilization under NIR irradiation. Reproduced with permission [39].

**Figure 4 pharmaceutics-15-00068-f004:**
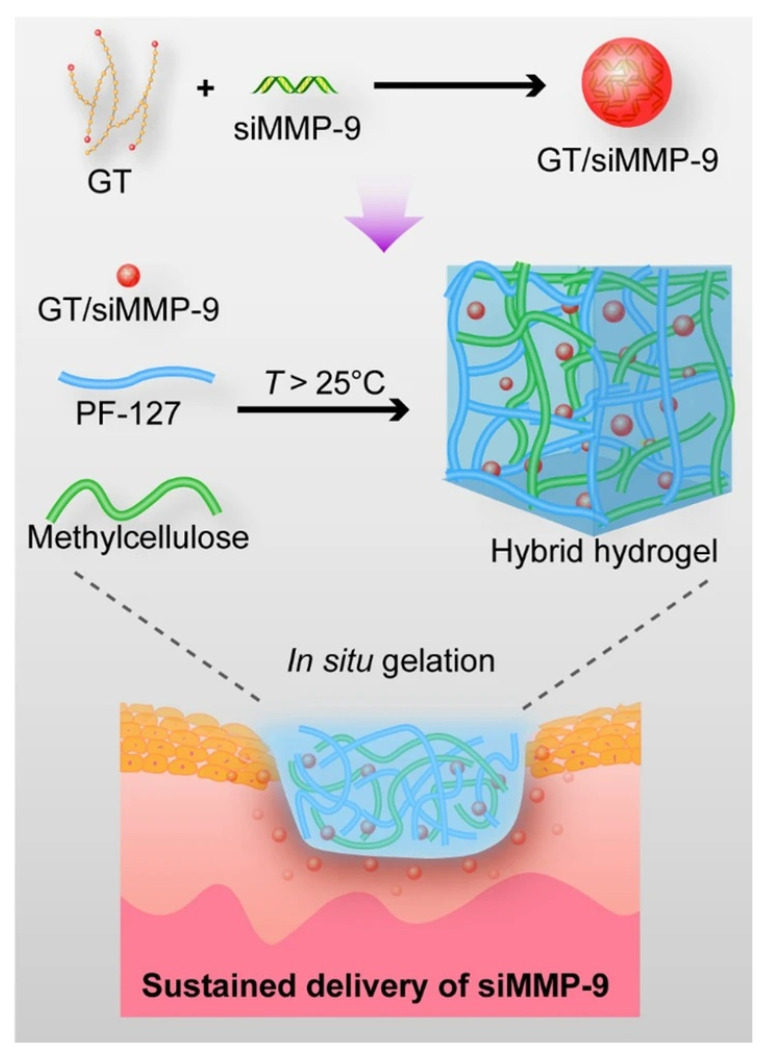
The schematic illustration of the preparation of the hybrid hydrogel dressing fabricated by Biyun Lan et al. Reproduced with permission [91].

**Table 1 pharmaceutics-15-00068-t001:** Wound dressings composed of nanomaterial loaded with hydrogels.

Main Polymer	Nanomaterial/Main Functional Components	Characteristics/Property	Ref.
CS sponge	Iturin-AgNPs	It shows more effective inhibition of bacterial infection and promotion of wound healing process and quality.	Liangfu Zhou et al. [34]
Agarose film	Nanoantimicrobial Cs_2.5_H_0.5_PW_12_O_40_	It can promote local acidic pH and exhibit a broad spectrum of biocidal activity with nonirritating acid levels for human skin models.	Roger H. Piva et al. [35]
Functionalized dialdehyde chitosan	Fe-MIL-88NH2 nanozyme	The enzyme-like activity and the reversible release of nanozymes influenced by pH make it can achieve intelligently adaptive trapping and killing of bacteria.	Yanyan Li et al. [36]
Poly(ε-caprolactone)-poly(ethylene glycol)-poly(ε-caprolactone) (PCEC) copolymer	PCEC-QAS nanoparticles	It promotes skin regeneration and prevents bacterial infection for MRSA-infected wound healing	Wenshuai Liu et al. [37]
hydrogel composed of adipic acid dihydrazide-modified γ-polyglutamic acid (γ-PGA-ADH) and aldehyde-(F127-CHO),	NO donor (N,N’-di-sec-butyl-N,N’-dinitroso-1,4-phenylenediamine, BNN6) loaded two-dimensional polydopamine nanosheets (PDA NS)	Under 808 nm irradiation, the embedded PDA NS exhibits outstanding photothermal transform properties and on-demand NO release. The combination of photothermal and NO gas therapy show an antibacterial effect.	Genhua Liu et al. [38]
PVA hydrogel	Novel gold cage (AuNCs) modified with epigallocatechin gallate (EGCG)	It has a high and stable photothermal conversion efficiency under near-infrared irradiation. The production of large amounts of ROS leads to the disruption of bacterial membranes, inducing bacterial lysis and apoptosis.	Jiaxin Ding et al. [39]
PCL hydrogel	Inorganic SPC salt (dressings were made using electrospinningtechnology)	It is capable of continuously generating oxygen for up to 10 days and cell studies further confirmed pronounced expression of HIF-1α at gene and protein levels.	Mubashra Zehra [40]
Hyaluronate gel	Nano-oxygenated (NOX) powder	It can deliver dissolved oxygen locally into the wound surface and only relieve hypoxic conditions without achieving excessive oxygen content causing hyperoxygenation damage to the tissue.	Zhengyang Yang [41]
Hydrophilic polytetrafluoroethylene(PTFE) membrane	hydrogel beads containing active SynechococcusElongatus (S. elongatus) PCC7942,	It can provide continuous dissolved oxygen to improve chronic wound healing and promote cell proliferation, migration, and tube formation in vitro.	Huanhuan Chen [42]
bacterial cellulose (BC, synthesized by Acetobacter xylinum)	hyperbranched cationic polysaccharide derivatives (HCP) encapsulating MMP-9 specific siRNA (siMMP-9)	The BC slowly released HCP/siMMP-9. The released siMMP-9 effectively reduced the gene expression and protein levels of MMP-9	Na Li [43]
Oxidized hydroxymethyl propyl cellulose (OHMPC) and adipic dihydrazide-modified hyaluronic acid (HA-ADH)	siRNA-29a gene-loading hyaluronic acid-polyethyleneimine complex HA-PEI@siRNA-29a	It can achieve downregulation of miR-29A by slow release of siRNA-29a and boost the wound healing process via the angiogenesis and type I collagen synthesis	Linglan Yang [44]
Complex hydrogels with chemically modified hyaluronic acid (HA), dextrose (Dex), and β-cyclodextrin (β-CD)	Resveratrol (Res) and vascular endothelial growth factor (VEGF) plasmids.	It accelerates the splinted excisional burn wound healing, particularly by inhibiting inflammation response and promoting microvascular formation while being biocompatible.	Peng Wang [45]
Zwitterionic hydrogels	Cerium oxide nanoparticles conjugated with miRNA146a	The hydrogel is injectable, self-healing, and with sustained release profiles. The sustained release of miRNA146a-tagged cerium oxide nanoparticles can speed up diabetic wound healing time and significantly reduce inflammation.	Gulsu Sener [46]
Gelatin and oxidized dextran	Nano-formulation of curcumin and cerium oxide	The hydrogel demonstrates a controlled and prolonged drug release, and accelerated cell migration besides providing a highly significant antioxidant and in-vivo anti-inflammatory activity	Syed Muntazir Andrabia [47]
Self-assembling peptidebased hydrogel	RADA 16-I for encapsulating PDGF-BB	the hydrogel can achieve the sustained release of PDGF-BB up to 48 h and show the angiogenic potential and wound healing ability of PDGF-BB	E. Santhinid [48]
Self-assembling peptidebased hydrogel	Human umbilical cord mesenchymal stem cells (hUC-MSCs) spheroids	It exhibits superior efficacy of faster healing by downregulating inflammatory factors to modulate the inflammatory response and upregulating VEGF to promote angiogenesis	Junshuai Xue [49]

## Data Availability

Not applicable.

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
