# Peer review of "Current Progress and Outlook of Nano-Based Hydrogel Dressings for Wound Healing"

_pharmaceutics, 2022, doi:10.3390/pharmaceutics15010068_

Round 1

Reviewer 1 Report

The paper is interesting because it makes a contribution to the field of hydrogels for wound dressings. There are plenty of schematics and references. 

Author Response

Response to Reviewer 1 Comments

Dear Reviewer,

Thank you very much for the comments from you. We have studied your comments carefully and have made revisions accordingly. All changes are marked up using the “Track Changes” function in the revised version. In addition, we have sent our paper to American Journal Experts for language polish. The following are the point-by-point responses to the comments. We expect that this revision will be satisfactory and could be accepted by pharmaceutics.

Thank you for your time and consideration.

Yours sincerely.

Hongwei Yao, MD, PhD

Professor of Surgery

Beijing Friendship Hospital

Capital Medical University

China

Email: yaohongwei@ccmu.edu.cn

Point 1: The paper is interesting because it makes a contribution to the field of hydrogels for wound dressings. There are plenty of schematics and references. 

Response 1: We are grateful for your interest and friendly comments on our manuscript. Your encouragement is a great motivation for us to keep moving forward. We made some modifications in the updated version to make the manuscript better and expect that this revision will be satisfactory and could be accepted by pharmaceutics.

On behalf of all the co-authors of this manuscript, we would like to express our great appreciation to the reviewer for your careful examination and constructive suggestion, again.

(the attachment contains the same content)

Reviewer 2 Report

The paper <Current progress and outlook of nanobased hydrogel dressings for wound healing>, proposed by Zhang et al. is an interesting one and could be published after major revision.

Reviewer 3 Report

The Authors have comprehensively reviewed the combination of hydrogels and nanomaterials for wound dressing to the purpose of chronic wound healing.

I find this review suitable for publication although few points should be addressed:

1) A more broad definition of hydrogel characteristics should be given in the Introduction by citing:

doi 10.3389/fbioe.2021.660145

doi 10.1002/adma.201601613

doi 10.1021/acsami.8b08874.

Just to male an example: the concept of "self-healing" is cited in the text but never explained.

Moreover, may you add a paragraph on how hydrogels work and at which temperatures.

2) Lines 75-98: a Table here is necessary to have an instant look at all the nanodelivery systems and their combination with hydrogels composed of various materials.

3) Lines 419-422: this a poor representation of the combinations of MSCs and extracellular vesicles derived therof. This is a new frontier which should be given appropriate relevance. Thus, expand these few lines with more examples and citing other references (e.g. doi 10.32604/biocell.2022.019448).

In the Introduction, define "chronic" as compared to "acute" wounds.

Minor concerns:

- Lines 4-5: something wrong with the authors list ("and"??)

- Line 37. do not repeat the second "skin"

- Lines 41-44: try not to repeat too many times "wound" and what is a "sassafras wound"? As far as I know this pant's leaves are used to cure wounds.

- Line 75: the correct sentence is "smaller than 100 nm"

- Lines 136-137: Actually, CS promotes haemostasi by inducing red blood cells aggregation. Please correct.

-Lines 273-275: Explain thr advantage of using exosomes.

Line 288: AM: please define.

Line 354: it should be miR-29a.

Discussion: I would introduce the issue of how to balance the ROS increase as antimicrobial with their reduction as a damaging agent.

Round 2

Reviewer 2 Report

The authors of the paper <Current progress and outlook of nanobased hydrogel dressings for wound healing> have revised this paper in according with all the requests and I appreciate their effort.

But, at line 159 <nanocellulose> was replaced by <nitrocellulose> and because between the two substances are great differences, I presume that nitrocellulose is not correct, but because I can not verify all the references I advise the authors to be sure of the correct form between nanocellulose and nitrocellulose.

At Line 167/ [39] is twice, and I propose to erase the first mention.  

<Some scholars believe that [39] AgNPs can induce the expression of free radicals and thus oxidize the outer membrane of the bacterium, which in turn causes 168 bacterial lysis and death [39].>
